# Effect of aberrant fructose metabolism following SARS-CoV-2 infection on colorectal cancer patients' poor prognosis

**Jiaxin Jiang** [1,☉,‡], **Xiaona Meng** [2,☉,‡], **Yibo Wang** [3], **Ziqian Zhuang** [3], **Ting Du** [4], **Jing Yan** [1]*

**1** Department of Biochemistry & Molecular Biology, China Medical University, Shenyang, China, **2** Teaching Center for Basic Medical Experiment, China Medical University, Shenyang, China, **3** Department of Bioinformatics, China Medical University, Shenyang, China, **4** Department of Pharmacology, School of Pharmacy, China Medical University, Shenyang, China

☉ These authors contributed equally to this work.
‡ These authors share first authorship on this work.
* jyan@cmu.edu.cn

**Data Availability Statement:** All human data, statistical analysis code, stimuli, network training sets, and trained networks are available on Zenodo at link http://doi.org/10.5281/zenodo.13358782.

## Abstract

Most COVID-19 patients have a positive prognosis, but patients with additional underlying diseases are more likely to have severe illness and increased fatality rates. Numerous studies indicate that cancer patients are more prone to contract SARS-CoV-2 and develop severe COVID-19 or even dying. In the recent transcriptome investigations, it is demonstrated that the fructose metabolism is altered in patients with SARS-CoV-2 infection. However, cancer cells can use fructose as an extra source of energy for growth and metastasis. Furthermore, enhanced living conditions have resulted in a notable rise in fructose consumption in individuals' daily dietary habits. We therefore hypothesize that the poor prognosis of cancer patients caused by SARS-CoV-2 may therefore be mediated through fructose metabolism. Using CRC cases from four distinct cohorts, we built and validated a predictive model based on SARS-CoV-2 producing fructose metabolic anomalies by coupling Cox univariate regression and lasso regression feature selection algorithms to identify hallmark genes in colorectal cancer. We also developed a composite prognostic nomogram to improve clinical practice by integrating the characteristics of aberrant fructose metabolism produced by this novel coronavirus with age and tumor stage. To obtain the genes with the greatest potential prognostic values, LASSO regression analysis was performed, In the TCGA training cohort, patients were randomly separated into training and validation sets in the ratio of 4: 1, and the best risk score value for each sample was acquired by lasso regression analysis for further analysis, and the fifteen genes *CLEC4A*, *FDFT1*, *CTNNB1*, *GPI*, *PMM2*, *PTPRD*, *IL7*, *ALDH3B1*, *AASS*, *AOC3*, *SEPINE1*, *PFKFB1*, *FTCD*, *TIMP1* and *GATM* were finally selected. In order to validate the model's accuracy, ROC curve analysis was performed on an external dataset, and the results indicated that the model had a high predictive power for the prognosis prediction of patients. Our study provides a theoretical foundation for the future targeted regulation of fructose metabolism in colorectal cancer patients, while simultaneously optimizing dietary guidance and therapeutic care for colorectal cancer patients in the context of the COVID-19 pandemic.

**Funding:** This work was partly supported by National Natural Science Foundation of China (81602510 to JY), High-Quality Development Project at China Medical University, supported by the Liaoning Provincial Department of Science and Technology (2023JH2/20200100 to JY) and National Natural Science Foundation of China (82370506). The funders had no role in study design, data collection and analysis, decision to publish, or preparation of the manuscript.

**Competing interests:** The authors have declared that no competing interests exist.

## Author summary

While diagnostic testing enables speedy detection of COVID-19 cases, the paucity of indicators for post-diagnostic risk assessment, particularly in cancer patients, hinders decisions for therapy allocation and cascade. The combination of clinical data with a vast array of omics technologies enables the development of new methods for describing complex patient metabolism and its relationship to clinical outcomes, particularly fructose metabolism, which has been shown to be closely associated with the progression of colorectal cancer. Although indicative of adverse outcomes in cancer patients, biomarkers of aberrant fructose metabolism may indicate systemic metabolic changes in COVID-19 patients. Moreover, the development of prognostic models of fructose metabolism showed COVID-19-specific prognostic indicators and disease-specific pathways of CRC, including aberrant fructose metabolism. In conclusion, targeting fructose metabolism is a novel therapeutic strategy that should be investigated in COVID-19 patients with CRC. This study aimed to investigate the differential expression and clinical significance of fructose metabolism-related genes in CRC patients infected with SARS-CoV-2, to investigate the effect of fructose metabolism changes on CRC development, to provide theoretical basis for further targeted regulation of fructose metabolism in CRC patients, and to help optimize the dietary guidance and clinical management of these patients.

## 1 Introduction

As of November 30, 2022, WHO had received reports of 639 million confirmed cases of Corona Virus Disease 2019 (COVID-19), including 6.6 million deaths (https://covid19.who.int/). Numerous clinical studies indicate that individuals with underlying disease are more prone to develop severe COVID-19 [1,2]. According to research, cancer patients are more prone to contract the severe acute respiratory syndrome corona virus 2 (SARS-CoV-2) and have a higher chance of becoming seriously unwell or even dying [3]. This could be attributed to several causes, including tumor burden, poor immunity due to malnutrition-induced [4,5] and even immunosuppressive therapy [6]. SARS-CoV-2 is more prevalent among cancer patients treated with anticancer therapy than those without anticancer therapy [7], this could be attributed to the potential for exposure within the hospital setting. Furthermore, the mortality rate of infected cancer patients was substantially higher than that of non-cancer patients [8,9]. As a result of the necessity for the majority of health care systems to rebuild their infrastructure and personnel in response to the COVID-19 pandemic, cancer care has encountered a number of problems throughout the epidemic [10,11]. Once cancer treatment has been canceled or suspended, rescheduling may encounter greater resistance [12]. The risk of SARS-CoV-2 infection and the incidence of severe disease vary across cancer types [13]. A multicenter retrospective observational study of patients with novel coronavirus and cancer discovered that patients with gastrointestinal cancer were more likely to progress to severe disease [14]. Worldwide, colorectal cancer (CRC) also has the fourth-highest incidence, with more than 1.9 million cases diagnosed each year [15]. The emerging coronavirus pandemic presents a formidable obstacle for CRC screening and diagnostic processes [12,16]. During the pandemic, the ability of colon surgeons to treat CRC patients was impaired [16,17], resulting from a halving of the diagnostic yield for digestive system cancers due to the delayed endoscopic screening in the majority of countries [18]. Delays in endoscopic diagnosis would increase the risk of CRC progression and metastasis, and the influence remains irreversible

even if screening is resumed. Certain malignancies that necessitate invasive surgery, such as colorectal and stomach cancers, are forced to choose different treatment plan as a result of measures made to prevent the spread of the disease and the overload of health services world-wide [19]. Comparing 25,666 patients treated for gastrointestinal cancer in 2020 with 23,530 individuals followed in 2019, it is found that a statistically significant decrease in the amount of radiation treatment and surgical treatment for patients [20]. According to another study, the number of surgical procedures for colorectal cancer decreased by around 15% ($P = 0.04$) [21]. The statistical data imply that this epidemic has a detrimental effect on the management of individuals with various gastrointestinal cancers. Additionally, investigations have empha-sized the particularities of the efficacy of existing immunizing drugs in the oncology group [22–24]. In general, it is crucial that the influence of SARS-CoV-2 on the prognosis of CRC patients in the context of the current COVID-19 pandemic.

COVID-19, caused by SARS-CoV-2 infection, has rapidly spread around the globe. This extremely pathogenic RNA virus is composed of structural proteins, including spike proteins, membrane proteins, nucleocapsid proteins, and envelope proteins [25,26]. The Angiotensin-converting enzyme-2 (ACE2) cell receptor is a key element in viral pathogenesis, as it facilitates the process of SARS-CoV-2 entry into susceptible cell. The receptor binding domain of the spike protein binds to the ACE2 receptor and triggers fusion of the virus with the host cell membrane, viral RNA is then released into the cytoplasm, and infection is established [27]. In addition to the lung, ACE2 receptors are also expressed in the kidney and gastrointestinal tract [28], including ileal and colonic enterocytes [29]. Although the primary target of SARS-CoV-2 is the respiratory system, it is capable of causing organ failures in various systems [30]. The intestines are targets for SARS-CoV-2 infections because some infected patients experience vomiting and diarrhea [31]. Furthermore, novel coronavirus ribonucleic acid is found in stool samples from infected persons [32]. Attachment of viral proteins to host cell receptors is a cru-cial step in SARS-CoV-2 infection. Due to the strong affinity between the human ACE2 recep-tor and the SARS-CoV-2 spike protein, the viral envelope and host cell membrane fuse to mediate the entry of SARS-CoV-2 into cells [33]. It has been demonstrated that dietary intake can affect the function and expression of the ACE2 gene [34,35], and daily fructose-rich diet in particular have a significant effect on ACE2 protein levels [36]. This indicates that high intake of fructose may be a risk factor for SARS-CoV-2 infection.

In order to sustain their life cycle, viruses alter the metabolic processes of their host cells [37]. DNA and RNA viruses impact many crucial carbon metabolic processes in the host, including glycolysis, pentose phosphate activity, and the synthesis of nucleotides, amino acids, and lipids [38]. Despite some viruses increase the demand for essential nutritional components and glutamine and combine critical metabolic pathways for anabolism, the precise metabolic changes generated by specific viruses tend to vary [38]. During SARS-CoV-2 infection, fruc-tose levels are elevated in human peripheral blood mononuclear cells (PBMCs). They are high in PBMCs from mild patients, moderate in those from moderate patients, and low in healthy controls and convalescent patients [39]. Previous research has linked fructose metabolism to the inflammatory pathways that produce IL-1β and IL-6 [40]. In the recent transcriptome investigations, it is demonstrated that the fructose metabolism is altered in patients with SARS-CoV-2 infection [41,42]. To make matters worse, high fructose intake may encourage tumor growth [43,44]. Therefore, the poor prognosis of cancer patients resulting from SARS-CoV-2 may be mediated by fructose metabolism.

In this study, we explored the expression and clinical significance of differential fructose metabolism-related genes in CRC patients infected with SARS-CoV-2. Using CRC cases from four distinct cohorts, we built and validated a prognostic model based on COVID-19 produc-ing fructose metabolic anomalies. We also developed a composite prognostic nomogram to

improve clinical practice by integrating the characteristics of aberrant fructose metabolism produced by this novel coronavirus with age and tumor stage. The association between this profile and clinical outcome, pathway activation, immunological heterogeneity, or medication response profiles was then investigated. Additionally, we constructed a ceRNA network to look for target miRNAs and lncRNAs of these prognostic mRNAs associated with SARS-CoV-2-induced fructose metabolism anomaly. In conclusion, our study provides a theoretical foundation for the future targeted regulation of fructose metabolism in CRC patients, while simultaneously optimizing dietary guidance and therapeutic care for CRC patients in the context of the COVID-19 pandemic.

## 2. Results

### 2.1. Screening Fructose Metabolism Differential Genes in COVID-19 Cohort

As a result of recent transcriptome discoveries indicating that patients infected with SARS-CoV-2 have an altered fructose metabolism [41,42], we initially screened novel coronavirus patients for differential genes associated with fructose metabolism. We downloaded GSE183533 data sets [45] linked with SARS-CoV-2 infection from the GEO database, which contains the expression profiling data for 41 samples, including 31 infected and 10 uninfected ones. The screened 6,155 differential genes are comprised of 3,768 up-regulated genes and 2,387 down-regulated genes (adj.$P$.Val $< 0.01$), as shown in the volcano plot and heat map (Fig 1A and 1B). Among them, there are 420 fructose metabolism-related genes (Fig 1C). We selected the 420 differential genes for further analysis.

### 2.2. Construction of CRC Prognostic Model of Differentially Expressed Genes Associated with Fructose Metabolism in Patients with COVID-19

The cross-discovery of these 420 fructose metabolism-related differential genes with the TCGA dataset and the two GEO datasets (GSE41258 [46,47], GSE17537 [48–51]) utilized as the external validation set led to the identification of 333 co-owned genes. Cox univariate regression analysis was used to screen 333 candidate genes for prognostic associated genes (S1 Table). Thirty genes with significant difference ($P < 0.05$) were used as prognostication-related genes for further analysis (Table 1). To further identify key genes in the candidate gene set, we gathered clinical data from COADREAD patients, and coupled Cox univariate regression and lasso regression feature selection algorithms to identify hallmark genes in colorectal cancer (Fig 2A and 2B). To obtain the genes with the greatest potential prognostic values, LASSO regression analysis was performed. In the TCGA training cohort, patients were randomly separated into training and validation sets in the ratio of 4:1. And the best risk score value for each sample was acquired by lasso regression analysis for further analysis. Fifteen genes were finally selected, including *CLEC4A*, *FDFT1*, *CTNNB1*, *GPI*, *PMM2*, *PTPRD*, *IL7*, *ALDH3B1*, *AASS*, *AOC3*, *SERPINE1*, *PFKFB1*, *FTCD*, *TIMP1* and *GATM* (Table 2).

**Risk Score** = -0.243521825861446×Exp*CLEC4A*-0.168666968006741×Exp*FDFT1*
-0.154439075167549×Exp*CTNNB1*-0.142519344319323×Exp*GPI*
-0.13556053647837×Exp*PMM2*-0.121053418829449×Exp*PTPRD*
-0.073494469859272×Exp*IL7*+0.00368615312167242×Exp*ALDH3B1*
+ 0.043305048523675×Exp*AASS*+0.0641212779356029×Exp*AOC3*
+0.0679577809044814×Exp*SERPINE1*
+0.121958919849076×Exp*PFKFB1*+0.130947608725393×Exp*FTCD*
+0.141053757356557×Exp*TIMP1*+0.25417517580763×Exp*GATM*

A

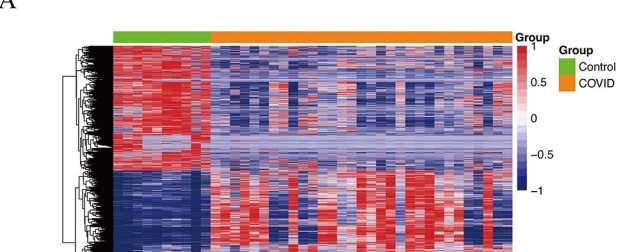

B

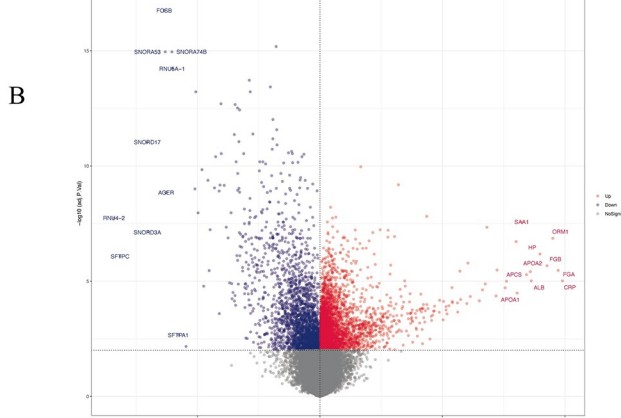

C

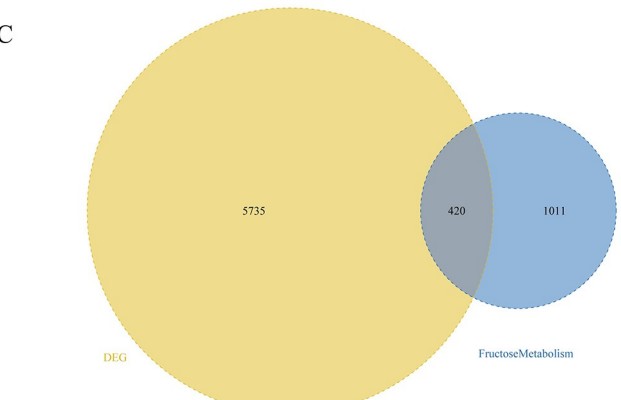

**Fig 1. Identification of differential genes associated with novel coronavirus fructose metabolism.** (A) Volcano plot of GSE183533 differential expression, blue indicates down-regulation of differential expression, red indicates up-regulation of differential expression, and differential gene screening condition is adj.*P*.Val < 0.01. (B) Differential gene expression heatmap. (C) Venn diagram of differential versus fructose metabolism genes.

**Table 1. Thirty prognostication-related genes for further analysis.**

| Gene | HR | Z | P value | Lower | Upper |
|------|-----|-----|---------|-------|-------|
| DLAT | 0.710999462 | -3.779482339 | 0.000157155 | 0.595732972 | 0.8485685 |
| PMM2 | 0.670847965 | -3.692514736 | 0.000222047 | 0.542746779 | 0.829184086 |
| TIMP1 | 1.423551734 | 3.498708447 | 0.000467517 | 1.168030289 | 1.734971737 |
| FDFT1 | 0.722140084 | -3.358560417 | 0.000783496 | 0.597195874 | 0.873224888 |
| CTNNB1 | 0.754347118 | -3.018853068 | 0.002537336 | 0.628181916 | 0.90585157 |
| AOC3 | 1.313686551 | 2.998538176 | 0.002712782 | 1.099109152 | 1.570155569 |
| CLU | 1.286604789 | 2.905767217 | 0.003663539 | 1.085484176 | 1.524989419 |
| FTCD | 1.105849777 | 2.628056308 | 0.00858743 | 1.025907558 | 1.192021366 |
| PFKFB1 | 1.153612816 | 2.627419111 | 0.008603529 | 1.036968108 | 1.283378455 |
| MAPK1 | 0.823847808 | -2.547071437 | 0.010863119 | 0.709727277 | 0.956318339 |
| ALDH3B1 | 1.275516517 | 2.485516505 | 0.012936359 | 1.052802482 | 1.545344368 |
| SLC25A24 | 0.80272543 | -2.365903759 | 0.017986119 | 0.669126689 | 0.962998676 |
| UQCRC2 | 0.804865608 | -2.356766942 | 0.018434813 | 0.671923548 | 0.964110648 |
| GNMT | 1.122755374 | 2.272877499 | 0.023033563 | 1.016068292 | 1.240644591 |
| HSPA9 | 0.80444463 | -2.257453132 | 0.023979775 | 0.665956885 | 0.97173132 |
| PTPRD | 0.800334413 | -2.254445978 | 0.024168126 | 0.659444011 | 0.971326089 |
| GATM | 1.197160827 | 2.232215289 | 0.025600737 | 1.0221905 | 1.402081165 |
| SERPINE1 | 1.219225293 | 2.228353932 | 0.025856923 | 1.024160974 | 1.45144206 |
| IGF1 | 1.202600594 | 2.223755196 | 0.026164919 | 1.022125774 | 1.414941513 |
| IL7 | 0.796090744 | -2.197594324 | 0.027978028 | 0.649582426 | 0.975642885 |
| LCAT | 1.189459003 | 2.182946966 | 0.029039716 | 1.017880454 | 1.38995961 |
| COG2 | 0.811006807 | -2.180361109 | 0.029230705 | 0.671807832 | 0.979047892 |
| STAT3 | 0.819194117 | -2.127009271 | 0.033419313 | 0.681672604 | 0.984459398 |
| CLEC4A | 0.694902519 | -2.123262046 | 0.033731899 | 0.49659833 | 0.972394551 |
| GPI | 0.819556045 | -2.078221297 | 0.037688981 | 0.679320877 | 0.98874057 |
| AASS | 1.212171664 | 2.048351913 | 0.04052553 | 1.008337352 | 1.457210863 |
| SDHB | 0.829373266 | -2.011671922 | 0.044254532 | 0.691175779 | 0.995202719 |
| HARS2 | 0.826649905 | -2.001422009 | 0.045346931 | 0.686050151 | 0.996064304 |
| DDAH1 | 0.821645281 | -1.996474443 | 0.045882305 | 0.677530635 | 0.996413937 |
| PLIN1 | 1.176309899 | 1.993451205 | 0.046212062 | 1.002731522 | 1.379935654 |

The median risk score was used to separate patients into high-risk and low-risk groups, which were then examined using Kaplan-Meier curves. In both training and test sets, high-risk OS was significantly lower than low-risk OS (Fig 2C and 2D). In addition, the ROC curve demonstrated that the AUC values of the training set and the test set for the three periods of 1 year, 3 years, and 5 years were greater than 0.70 (Fig 2E), indicating that the model exhibited a high level of validation efficacy.

The robust of the prognostic model was validated by utilizing external data sets. Data from COADREAD patients with survival data processed in the GEO database (GSE41258, GSE17537) was downloaded to predict the clinical classification of COADREAD patients in the GEO database according to the model, and kaplan-meier was used to assess the stability of the prediction model. In both GEO external validation sets, the results demonstrated that OS was significantly lower in the high-risk group than in the low-risk group (Fig 2F). In order to validate the model's accuracy, ROC curve analysis was performed on an external dataset, and the results indicated that the model had a high predictive power for the prognosis prediction of patients. AUC values counted in the three periods of 1 year, 3 years, and 5 years in GEO

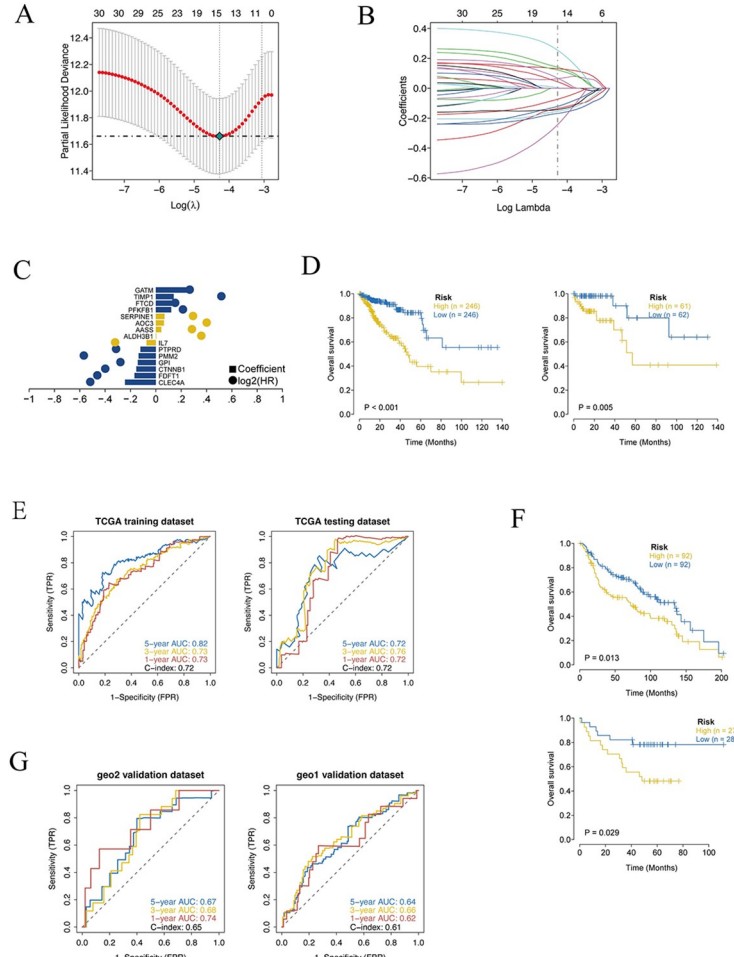

**Fig 2. Prognostic model of fructose metabolism genes in colorectal cancer.** (A) Ten cross-validations of tuning parameter selection in the LASSO model to determine the minimum lambda value. (B) LASSO coefficient distribution of prognostic genes and gene combinations at the minimum lambda value. (C) Coefficient of the Lasso gene. (D) Survival curves for TCGA training set and test set models. (E) ROC curves (years 1-3-5) for the TCGA training set model and the test set model. (F) Survival curve of GEO external dataset model, the survival curve of patients in high risk group was significantly lower than that in low risk group. (G) ROC curves for GEO external dataset models (years 1-3-5).

validation set 1 (GSE41258) were greater than 0.60, and AUC values counted in the three periods of 1 year, 3 years, and 5 years in GEO validation set 2 (GSE17537) (Fig 2G).

## 2.3. Immune Infiltration and Drug Resistance: Clinical Predictive Value of Models

The tumor microenvironment has a substantial impact on the diagnosis, survival result, and sensitivity to clinical treatment. Tumor-infiltrating immune cells' characteristics are increasingly linked to the onset and advancement of cancer [52]. The composition and concentration of tumor-infiltrating immune cells can predict patient survival and impact how tumors respond to treatment [53–55]. It is a remarkable fact that dietary stress exacerbates the metabolic tug of war in tumors and directly affects local CD8[+] T cell function [56]. To kill tumor cells, CD8[+] T cells require direct cell-to-cell contact and sufficient metabolic resources [56]. CD8[+] T cells rely on many of the same fuel sources and metabolic pathways as tumor cells to

**Table 2. 15 genes involved in the best risk score value for each sample.**

| Gene | Coef | HR | Lower | Upper |
|------|------|-----|-------|-------|
| CLEC4A | -0.243521826 | 0.694902519 | 0.49659833 | 0.972394551 |
| FDFT1 | -0.168666968 | 0.722140084 | 0.597195874 | 0.873224888 |
| CTNNB1 | -0.154439075 | 0.754347118 | 0.628181916 | 0.90585157 |
| GPI | -0.142519344 | 0.819556045 | 0.679320877 | 0.98874057 |
| PMM2 | -0.135560536 | 0.670847965 | 0.542746779 | 0.829184086 |
| PTPRD | -0.121053419 | 0.800334413 | 0.659444011 | 0.971326089 |
| IL7 | -0.07349447 | 0.796090744 | 0.649582426 | 0.975642885 |
| ALDH3B1 | 0.003686153 | 1.275516517 | 1.052802482 | 1.545344368 |
| AASS | 0.043305049 | 1.212171664 | 1.008337352 | 1.457210863 |
| AOC3 | 0.064121278 | 1.313686551 | 1.099109152 | 1.570155569 |
| SERPINE1 | 0.067957781 | 1.219225293 | 1.024160974 | 1.45144206 |
| PFKFB1 | 0.12195892 | 1.153612816 | 1.036968108 | 1.283378455 |
| FTCD | 0.130947609 | 1.105849777 | 1.025907558 | 1.192021366 |
| TIMP1 | 0.141053757 | 1.423551734 | 1.168030289 | 1.734971737 |
| GATM | 0.254175176 | 1.197160827 | 1.0221905 | 1.402081165 |

support proliferation, survival, and effector functions [56]. Several recent studies have found that eating disorders alter systemic metabolic status and impact anti-tumor immunity [57–59]. The probable biological mechanisms by which risk score influences the progression of colorectal cancer were further investigated by studying the correlation between risk score and tumor immune infiltration. The results revealed the proportion of immune cells in each subject (Fig 3A). There were numerous correlations between immune factors (Fig 3B). In addition, the levels of immune factors such as T cells CD4$^+$ memory resting and T cells CD4$^+$ memory activated were significantly greater in the low-risk group compared to the high-risk group, although the levels of immune factors such as B cells naive and Macrophages M0 were significantly lower (Fig 3C). Risk scores were considerably positively connected with T cells regulatory (Tregs), B cells naive, and macrophages M0, and significantly negatively correlated with T cells CD4$^+$ memory activated, T cells CD4$^+$ memory resting, and Eosinophils (Fig 3D).

The efficacy of early colorectal cancer surgery paired with chemotherapy is evident. We propose assessing combinations of individual immune determinants or immune determinants to improve patient evaluation for predicting response to therapy or resistance, based on compelling data supporting their prognostic and predictive capabilities. We utilized the Rpackage "pRRophetic" to predict the chemosensitivity of each tumor sample based on drug sensitivity data from the GDSC database and then explored the risk score's sensitivity to common chemotherapeutic drugs. The results demonstrated that the risk score was substantially linked with the susceptibility of patients to bleomycin, camptothecin, cisplatin, cytarabine, doxorubicin, gemcitabine, and other medicines (Fig 3E). From the TISIDB database, we extracted correlations between these important genes and various immune factors, including immunomodulators, chemokines, and cellular receptors (Fig 3F). All data revealed that these 30 critical genes (Table 1) have a significant impact on the immune microenvironment, and are closely associated with the level of immune cell infiltration.

## 2.4. Correlation Analysis between Disease Risk and Multiple Clinical Indicators

We integrated the clinical information and risk scores of patients in the high and low risk groups and presented their results through regression analysis as nomograms. The results of

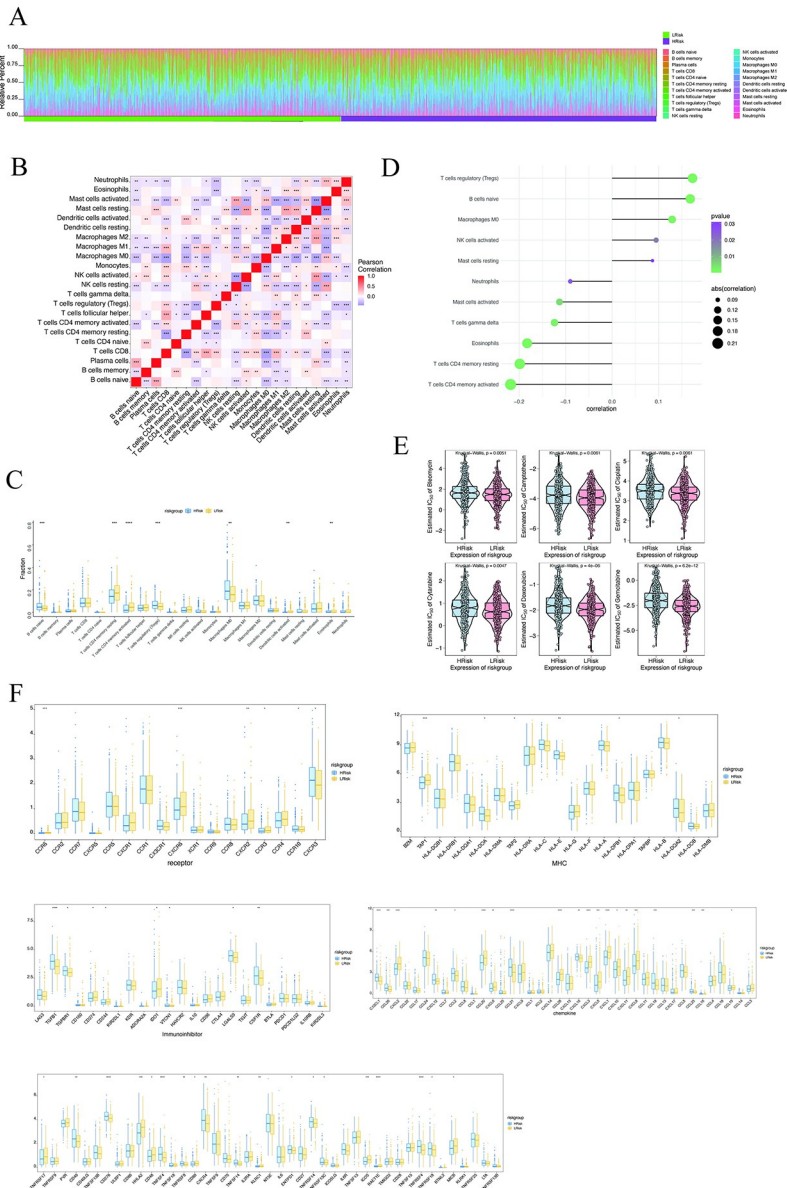

**Fig 3. Multi-omics study to investigate the clinical predictive value of the model.** (A) Relative percentages of 22 immune cell subsets in high-risk and low-risk patients. (B) Pearson correlation between 22 immune cells, blue indicates positive correlation and red indicates positive correlation. (C) Differences in immune cell content between high-risk and low-risk patients, with blue indicating low-risk patients and yellow indicating high-risk patients. (D) Association of risk score with immune cells. (E) Sensitivity analysis of risk score versus common chemotherapeutic agents. (F) Differential expression of chemokines, immunostimulants, MHC, receptors, and immunosuppressive agents between patients in the high- and low-risk groups, with blue indicating low-risk patients and yellow indicating high-risk patients.

logistic regression analysis demonstrated that the distribution of age, stage, T, and risk score values of colorectal cancer clinical indicators contributed to multiple scoring processes in all of our samples. The predictive analysis scoring procedure was influenced by the distribution of risk score values (Fig 4A). Simultaneously, we conducted prognostic analyses for the three-year and five-year periods of colorectal cancer, and the results indicated that the model had a high predictive power for the prognosis prediction of patients (Fig 4B). Univariate and

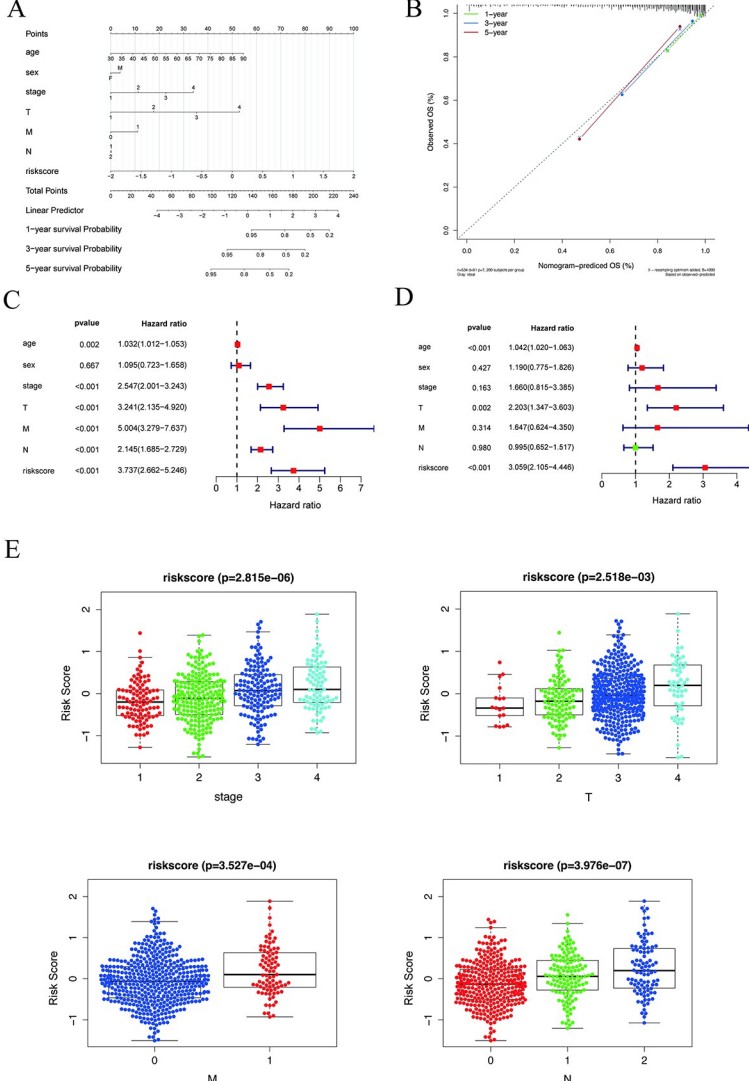

**Fig 4. Construction and efficacy evaluation of nomogram model related to risk score.** (A) Nomogram associated with the risk model. (B) Model-dependent correction curve. (C-D) Forest plots of univariate regression and multivariate regression, with red indicating risk factors and green indicating protective factors. (E) Clinical relevance of risk score for colorectal cancer. Risk scores differed from tumor stage, T, M, N, age, sex and $P < 0.05$ was considered statistically significant.

multivariate analyses revealed that the risk score is also an independent prognostic factor in COADREAD patients (Fig 4C and 4D).

Subsequently, we grouped the risk score values of all samples by different clinical indicators, presented them as boxplots, applied the rank sum test (When there are only two indicator classification values (age, sex, M) use the t-test. The Kruskal-test was used for other cases), and determined that these risk score values were significantly different between groups for stage, T, M, and N ($P < 0.05$) (Fig 4E).

## 2.5. Specific Signaling Mechanisms Associated with Prognostic Models

We studied the precise signaling pathways engaged in the high and low risk-related models to investigate the potential molecular mechanisms by which risk scores influence tumor growth.

GSEA analysis revealed significant enrichment in many related pathways, of which GO enriched pathways were chondrocyte differentiation, chromosome separation, detection of temperature stimulus. KEGG enriched pathways were aminoacyl trna biosynthesis, axon guidsis, dilated cardiomyopathy. We present several of these highly significant pathways separately in a centralized fashion (Fig 5A and 5B), suggesting that perturbation of these signaling pathways in patients at high and low risk impacts the prognosis of colorectal cancer patients.

Using these 15 genetic markers from the gene set (Fig 2C), we discovered that they are regulated by common mechanisms, such as multiple transcription factors. Consequently, these transcription factors were enriched utilizing cumulative recovery curves (Fig 5C and 5D). Annotation of motif-TF as well as the selection of essential genes, the investigation revealed that cisbp_M6345 was the motif with the highest normalized enrichment score (NES: 5.22). *ALDH3B1*, *FTCD*, *PFKFB1*, and *PTPRD*, were enriched in this motif. We present all enriched motifs and transcription factors for the modelled genes (S2 Table).

15 modeling genes were analyzed to determine their potential miRNAs and lncRNAs using miRWalk and ENCORI databases, respectively. First, 15 mRNA-miRNA relationship pairs linked with mRNAswere retrieved from the miRWalk database (S3 Table, 946 miRNAs in total), and only 14 mRNAs and 33 miRNAs associated with disease miRNAs were kept (Fig 5E). Interacting lncRNAs were then predicted based on these miRNAs. A total of 2,926 pairs of interactions were obtained (S4 Table, 10 miRNAs and 1,868 lncRNAs), and a ceRNA network was created using cytoscape (v3.7) (Fig 5F). In addition, enrichment analysis on 14 mRNAs was ran, and Metascape revealed that the genes were primarily enriched in the small molecule biosynthetic process, the negative regulation of endopeptidase activity, the small molecule catabolic process, and the cell activation pathway (Fig 5G).

## 2.6. Expression of modeled genes in CRC tissues

To further assess the carcinogenesis of model genes in CRC, we analyzed their expression. The RNA expression data of patients with SARS-CoV-2 infections, as well as normal and malignant tissue samples, were collected from public sources. First, GEO data sets GSE183533 demonstrated that COVID-19 patients had considerably greater expression levels of *SERPINE1*, *PMM2*, *GPI*, *FTCD*, and *TIMP1* than controls (Fig 6A). Furthermore, these five proteins exhibited considerably higher expression levels in CRC tumor tissues than in normal tissues based on the Human Protein Atlas Project (Fig 6B).

## 3. Discussion

Many viruses possess similarities, such as invasion, proliferation, and spread within host cells [60]. Currently, disruptions in the regulation of host metabolic processes, or virus-induced metabolic reprogramming, provide insight into the pathogenesis of viral infections [61]. In addition to infection and severe condition, it has been shown that SARS-CoV-2 hosts have a significant persistent metabolic consumption after rehabilitation [62]. The COVID-19 epidemic poses a significant challenge to global public health, and its impact on human metabolism is particularly complex and profound. Metabolic abnormalities have been extensively investigated in tumor development and treatment, and novel coronavirus infections may further exacerbate these problems. The coronavirus itself and its treatment may lead to metabolic dysfunction in cancer patients. Inflammatory and immune responses triggered by viral infections, as well as the use of antiviral drugs, may affect the body's energy metabolism, glucose homeostasis and lipid metabolism. These changes may not only increase energy expenditure in cancer patients, but may also affect the pharmacokinetics and therapeutic effects of the drug. Second, lifestyle changes and psychological stress during the novel coronavirus epidemic

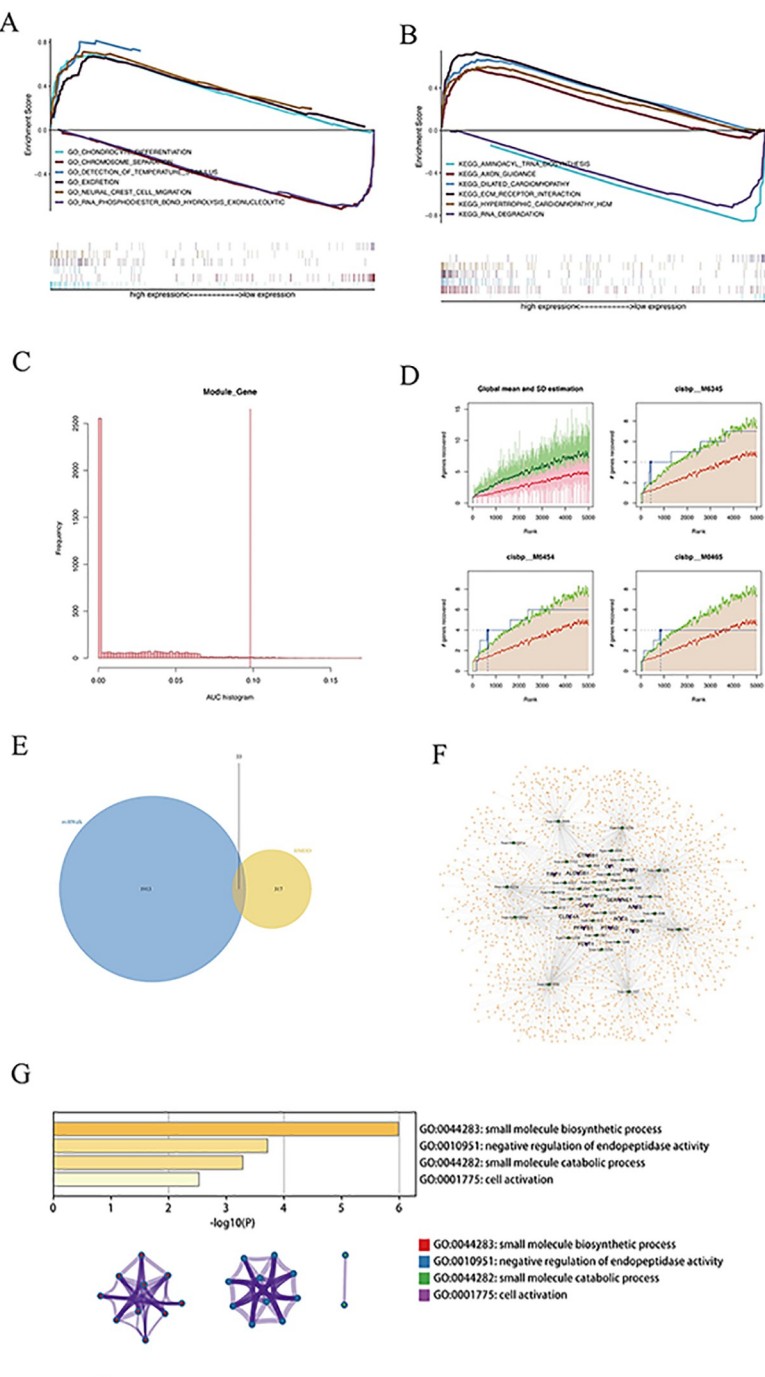

**Fig 5. Enrichment analysis of risk scores and transcriptional regulation analysis of model genes.** (A-B) GO and KEGG enrichment analysis of risk scores. (C) Distribution of AUC values of enriched motifs, which are calculated from recovery curves of motif ranking by core genes. (D) For the four motifs with higher AUCs, the red line in the figure is the mean of the recovery curves for each motif, the green line is the mean ± SD (standard deviation), and the blue line is the recovery curve of the current motif. The maximum distance point (mean ± SD) between the current motif and the green curve is the maximum enrichment rank selected. (E) Wayne plot of miRNAs predicted by miRwalk and HMDD databases. (F) ceRNA network of model genes, purple indicates mRNA, green indicates miRNA, and orange indicates lncRNA. (G) GO-KEGG enrichment analysis based on Metascape.

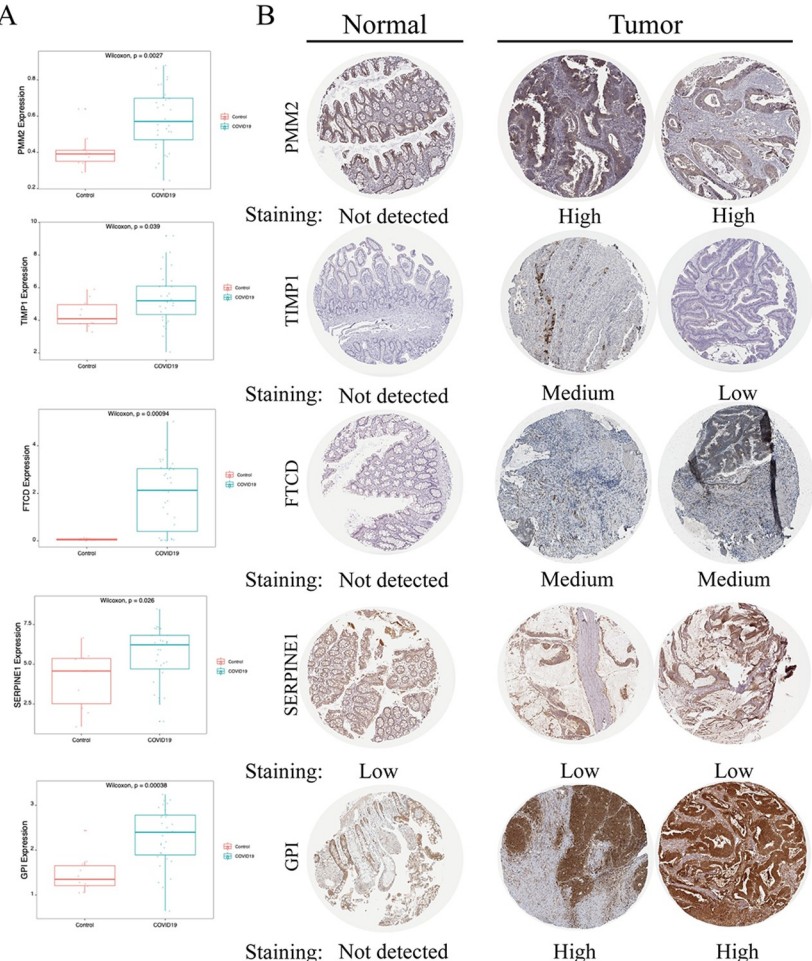

**Fig 6. Expression of model genes in CRC tissues.** (A) Differential expression of model genes in COVID-19 patients, red is the control group and blue is the COVID-19 patient group. (B) Human Protein Atlas project.

may further exacerbate metabolic problems in cancer patients. For example, decreased exercise and changes in eating habits caused by lockdown measures may aggravate obesity and malnutrition, which are closely related to tumor development and treatment outcomes. In addition, the interaction between novel coronavirus infection and the tumor itself also needs to be studied in depth, and future studies need to deeply investigate the mechanisms of these effects and develop individualized management strategies to optimize the health and treatment outcomes of cancer patients.

For lack of metabolic functions, viruses have to rely on the host cells to receive the necessary components for replication [63]. Therefore, the majority of viruses drastically affect the metabolism of the host cells. And some eventually stop all cellular metabolism to lead to cell death [63], which is also the typical fate of SARS-CoV-2-infected cells. After infection, the respiratory epithelial cells must be replaced to prevent the lungs losing their functions, including gas exchange and the synthesis of surfactants essential for lung function [64]. On the basis of the existing research about metabolic consequences *in vivo*, those patients infected with SARS-CoV-2 may experience higher blood glucose, fatty acid concentrations, as well as aberrant amino acid metabolism. Through using single-cell RNA sequencing, researchers discovered a significant rise in glycolytic metabolites, HIF-1α and genes related to oxidative stress in the

bronchoalveolar cells of patients with COVID-19 [65,66]. It was revealed that more energy is supplied by glycolysis in infected cells, and many components of normal metabolism may be affected during the progression of the disease. This offers us the therapeutic targets which aim to restore metabolic balance. Restoring metabolic balance in patients with COVID-19 is a complex and important topic involving multiple aspects of disease development, treatment strategies, and patient rehabilitation. The effects of restoring metabolic balance on patients with COVID-19 are multifaceted and multifaceted, and it is necessary to comprehensively consider pathophysiological processes, individualization of treatment strategies, and the need for long-term rehabilitation. Future research and clinical practice should further explore how to maximize the recovery of metabolic balance in patients by optimizing nutritional support and treatment strategies, thereby improving treatment outcomes and prognosis.

Modern lifestyles, including sleep deprivation, daily stress, and imbalanced diets, can result in chronic low-grade inflammation and damage to the immune system [67]. Sugar is frequently added to beverages and foods in the form of sucrose or industrial goods such as high-fructose corn syrup (HFCS). HFCS55, consisting of 55% fructose and 45% glucose, is the most prevalent formulation in beverages. Sucrose needs to be broken down into fructose and glucose before absorption. Fructose absorption is very variable, based on age, sugar intake, and other unknown genetic, nutritional, and environmental variables [68]. Reducing fructose intake may aid in the treatment of numerous gastrointestinal disorders [69]. Compared to glucose, the association between fructose and cancer cell metabolism has been disregarded for a considerable amount of time. To adapt to various situation, the metabolic plasticity of cancer cells enables them to exploit any available carbon source. In recent years, the accumulated experimental and clinical data has suggested that greater fructose consumption is associated with the development of malignancies. Even at moderate levels, fructose has been reported to stimulate fatty acid production and lead to oncogenesis in mice [70]. Fructose is transported intracellularly mainly through Glucose Transporter 5 (GLUT5, also known as SLC2A5) with a high affinity for fructose ($K_m$ = 6.0 mM), which is strongly expressed at the apical end of the small intestine luminal membrane [71]. Multiple studies have discovered that GLUT5 is expressed in human colorectal epithelial adenocarcinoma cells [72,73]. Dietary fructose intake has been suggested as a potential cancer driver because fructose provides an advantage for cancer cells malignant proliferation. In human colorectal tumors, GLUT5 is expressed at higher levels compared with adjacent normal intestinal epithelial cells [74], and similarly, other enzymes that metabolize fructose, such as ketohexokinase and aldolase B, have also been reported [75,76]. Ingestion of as little as 5 g of fructose has been reported to cause GLUT5 saturation (i.e., malabsorption) in the small intestine, resulting in increased fructose concentrations in the intestinal lumen of healthy individuals [77–79]. A study in mice showed that when fructose doses exceed 1 g/kg, fructose absorption in the small intestine is weakened, resulting in increased fructose concentrations in the large intestine [80]. Moreover, fructose can be captured by tumors in the intestinal lumen rather than being delivered to the liver and blood [81]. These results suggest that fructose concentrations are higher in the intestinal lumen after oral administration of HFCS and that intestinal tumors can transport fructose directly from the intestinal lumen. The impact of direct fructose absorption by intestinal tumors in the intestinal lumen is a complex topic involving multiple aspects of intestinal physiology, tumor biology, and glucose metabolism. This process is closely related to the special metabolic needs of tumor cells and intestinal glucose metabolism pathways, providing a theoretical basis for further exploration of the nutritional needs and therapeutic targets of intestinal tumors.

In addition to CRC, fructose metabolism is also enhanced obviously in other types of cancer, and is closely associated with poor prognosis. Under fructose-rich circumstances, transketolase flux along the nonoxidative pentose phosphate pathway is increased in pancreatic

cancer cells, resulting in preferential use of fructose for nucleotide synthesis [82]. GLUT5 was also highly increased in non-small cell lung cancer tissue relative to normal lung tissue, which promotes fructose metabolism to increase cancer cell proliferation in glucose-limited environments [83]. Aldolase B, a key enzyme in fructose metabolism, is increased due to high-fructose diet, which may accelerate liver metastasis of breast cancer and colon cancer [84]. In acute myeloid leukemia patients, GLUT5 gene expressionand fructose utilization are enhanced, which is correlated with worse clinical outcomes [85]. Fructokinase and GLUT5 are abundantly expressed in gliomas, which linked to glioma malignancy and poor patient survival [86]. All of these investigations indicate that cancer cells can use fructose as an extra source of energy for proliferation and metastasis [87]. In light of the glucose restriction in the tumor microenvironment, limiting fructose intake and inhibiting metabolism may be a promising cancer treatment strategy. A greater understanding of fructose metabolism in cancer is required for the development of better diagnostic and therapeutic schedule. The potential for targeted therapies for fructose metabolism is not limited to inhibition of proliferation of tumor cells, but involves modulation of the tumor microenvironment and immune responses. Fructose metabolites can affect tumor-related inflammatory response and the release of immuno-modulatory factors, and then affect the immune escape and metastasis ability of tumors. However, targeted therapies for fructose metabolism still face some challenges and unresolved issues. For example, metabolism of fructose in normal cells is also necessary, so targeted therapy needs to minimize adverse effects on normal cells while ensuring tumor cell-specific effects. In addition, the degree of dependence on fructose metabolism may vary between tumor types and subtypes, so treatment strategies need to be individualized and precise.

At present, sugar and fat in people's daily intake are gradually increased. The combination of these two nutrients exacerbates metabolic syndrome [88]. Based on the combined effects of fat and monosaccharide consumption, fructose, not glucose, exacerbated the negative effects of dietary fat [88–91]. Futhermore, fructose's pathogenic effect was more pronounced in mice on a high-fat diet [88]. Consequently, the addition of fructose is more detrimental than the addition of glucose, and the binding of fructose and glucose may be more detrimental than fructose alone [92]. Recent research has shown that dietary fat increases intestinal fructose metabolism and blood glycerol levels [93]. Sweet sensations and sugar-induced sensations may combine and interact, considerably boosting preferences for sugar-rich meals. It can lead to excessive consumption of fructose to induce excessive energy intake, overweight, and obesity. Fructose can additionally reduce meal-induced leptin compared with glucose and lead to leptin resistance, which is why fructose consumption is more prone to obesity [94,95]. Obese individuals are indeed more vulnerable to SARS-CoV-2 infection and exhibit more morbidity and mortality [96]. Their dulled immune response to SARS-CoV-2 originates in part from prior influenza virus experience [97]. Meanwhile, obesity is one of the contributing contributors to the worldwide increase in colorectal cancer incidence [98]. Gratifying research revealed that intermittent fasting can increase T cell antitumor responses [99]. In addition, fructose consumption may increase intestinal salt absorption to promote hypertension [100]. Hypertension has not only emerged as a significant risk factor for infection with the SARS-CoV-2 and subsequent illness severity [101], but it may also be a risk factor for malignancy [102]. In view of the above results, we should pay more attention to the diet management of CRC patients with COVID-19, especially fructose intake and nutrient combination. Diet management, particularly fructose intake, in patients with COVID-19 and colorectal cancer involves complex nutritional strategies and therapeutic considerations. Colorectal cancer, as a common malignant tumor of digestive system, requires special attention to nutritional support and immune function maintenance of patients during the epidemic. First, fructose, as a carbohydrate, should be carefully considered in dietary management during the new coronavirus

epidemic. The metabolic pathway of fructose may influence the growth and metabolic status of colorectal cancer cells, especially when inflammation and immune responses are activated. Therefore, controlling fructose intake may help to reduce the inflammatory response in patients and maintain a good nutritional status. Second, the impact of novel coronavirus infection on colorectal cancer patients may lead to changes in digestive function and loss of appetite, which further impacts nutrient intake and energy metabolism. Therefore, dietary management needs to be tailored to the individual patient's condition to ensure adequate calorie and nutrient supply while avoiding possible adverse effects of excessive fructose intake. In practice, colorectal cancer patients are recommended to prefer foods rich in dietary fiber and low fructose. These foods not only help maintain normal bowel function and digestive health, but also help control blood sugar and insulin responses and reduce the risk of inflammation. The health care team should develop appropriate dietary recommendations in conjunction with the patient's condition, nutritional status, and personal preferences, and continuously monitor and adjust to ensure that nutritional support and treatment outcomes are maximized.

A high-fructose diet has been linked to metabolic syndrome, also known as the "cardiometabolic disease syndrome" [103–106]. It is possible that additional physiological alterations caused by metabolic syndrome and type 2 diabetes (T2D) interact synergistically with COVID-19, ultimately aggravating the disease course. Both metabolic syndrome and T2D are chronic inflammatory diseases. Since the severe and critical stages of COVID-19 are driven by an exaggerated inflammatory response to infection (a cytokine storm), an elevated baseline inflammatory status in patients with pre-existing metabolic impairment may increase the likelihood of achieving pathogenic levels of the inflammatory response and physiological impairment [107]. The chronic inflammatory state of COVID-19 is mediated by pro-inflammatory cytokines, such as interleukin 6 and tumor necrosis factor alpha [108]. In fructose ingestion, these pro-inflammatory cytokines are also released from adipose tissue. Hence, their concentrations are elevated in individuals with metabolic disorders, in whom they will get an excessive inflammation. The relationship has been proven at the molecular and organ system levels [108,109]. In addition to initiate inflammatory reaction, fructose directly modifies immune cell metabolism and boosts cytokine responses during infection [110]. A recent analysis from 188 countries indicates that the consumption of sugar products has a significant effect on COVID-19 mortality [111]. In summary, reducing sugar intake may help to improve the immune response, reduce the inflammatory response and improve metabolic health of patients with COVID-19, thereby helping to improve the prognosis of patients with the disease. Therefore, during the COVID-19 epidemic, patients are advised to follow healthy eating habits and limit sugar intake to support the functioning and overall health of the immune system.

Our study found a correlation between risk score and tumor immune infiltration. COVID-19 can lead to abnormal immune system responses, including persistent inflammatory conditions and immunosuppression, all of which may create favorable conditions for tumor infiltration in colorectal cancer. By altering the intestinal immune environment and expression of inflammatory mediators, viral infection may directly or indirectly promote tumor growth and infiltration. Colorectal cancer, as a multifactorial disease, tends to develop under the combined influence of genetic, environmental, and metabolic factors, and COVID-19 may further increase the association between abnormal fructose metabolism and colorectal cancer invasion by exacerbating the interaction between these factors. Large-scale clinical and epidemiological data analysis of the relationship between abnormal fructose metabolism and colorectal cancer invasion after infection with SARS-CoV-2 is required to verify whether there is a significant association and to further explore its underlying molecular mechanisms. Although further clinical and experimental studies are required to validate these hypotheses, abnormal fructose

metabolism triggered by COVID-19 may influence colorectal cancer development and tumor infiltration through multiple mechanisms, which deserves our in-depth investigation in future studies. If abnormal fructose metabolism is associated with novel coronavirus infection, this may need to be taken into account in prevention and treatment strategies, such as reducing the risk of colorectal cancer through nutritional intervention, inflammatory control, or other means of metabolic regulation.

At the stage of virus infection, nothing is known about the extent of tumor growth and altered gene expression in tumor tissue. Inflammatory reactions in the lung, brain, heart, kidney, and vascular pathology are the focus of necropsy analyses [112]. Analysis of tissue biopsies for colorectal cancer patients infected with SARS-CoV-2 has yielded very little molecular data until now. Given the high hospitalization rate of CRC patients infected with new coronavirus, we may have a comprehensive understanding which therapy is more effective. The unfavorable consequences (including higher mortality) of hospitalized CRC patients paired with the novel coronavirus are partly resulted from lack of understanding of molecular etiology. Therefore, it is unclear that increased SARS-CoV-2 infectivity, respiratory distress syndrome, cellular or humoral immune dysfunction, coagulation disorders, cardiopulmonary impairment, vasculitis, especially metabolic disorders contribute to the clinical manifestations of severe COVID-19 in CRC patients. To develop an effective therapy for CRC patients, the potential molecular mechanisms is the crucial research objectives.

T helper (CD4$^+$) cells consist of different subsets such as T helper 1 (TH1) cells, T helper 2 (TH2) cells, T helper 17 (TH17) cells, regulatory T (Treg) cells, and T follicular helper (TFH) cells, each with diverse functions and varying prognostic significance [113]. The prognostic model showed a strong positive association between risk scores and Tregs. A study shown that the prognostic significance of Treg cells depends on their location, with intratumoral or circulatory Treg cells linked to lower overall survival and disease-free survival, but peritumoral Treg cells exhibited no such association [114]. A recent study shown various effects on survival depending on the spatial proximity of Treg cells and CD8$^+$ T cells in rectal cancer. Both relative interactions and placement in distinct compartments (epithelial versus mesenchymal) can have varying prognostic effects [115]. Regrettably, there is a scarcity of transcriptome data for patients infected with SARS-CoV-2, especially spatial information. Using spatial information in future studies can enhance prediction accuracy and potentially resolve contradictions seen in previous research.

Our study also found that risk scores were associated with sensitivity to several common chemotherapeutic agents.We speculate that COVID-19 may lead to increased release of inflammatory factors in the body, which in turn affects the metabolic status and drug sensitivity of tumor cells.Second, viral infections may alter the function of the immune system, thereby affecting the efficacy and toxic side effects of chemotherapeutic agents. In addition, the anti-tumor mechanism of some chemotherapeutic drugs is closely related to the metabolism of tumor cells, so we speculate that abnormal fructose metabolism may affect the mechanism of action of these drugs by changing the energy metabolism pathway in tumor cells. Common chemotherapeutic drugs, such as cisplatin, doxorubicin, and paclitaxel, inhibit the proliferation and metastasis of tumor cells through different mechanisms, however, the effectiveness and toxicity of these drugs may be affected by the metabolic status of the host.Abnormal fructose metabolism may alter drug metabolic pathways and drug clearance capacity in cancer patients, which in turn affect the concentration and bioavailability of chemotherapeutic drugs in the body.Abnormal fructose metabolism may also change the apoptotic pathway and DNA repair ability of tumor cells, which in turn affects the sensitivity of chemotherapeutic drugs.On the other hand, the COVID-19 epidemic may expose patients to more challenges in the treatment process, such as the timing of receiving chemotherapeutic drugs, adjustment of

treatment regimens, etc., and these factors may also affect the patient 'response to chemotherapeutic drugs.Based on the results of this study, treatment management strategies for cancer patients can be optimized, including regular monitoring of metabolic parameters, individualized adjustment of chemotherapy regimens, and early prevention and management of metabolic abnormalities to reduce the adverse effects of COVID-19 infection on patient treatment outcomes.Based on the understanding of metabolic abnormalities caused by COVID-19 infection, particularly focusing on the relationship between fructose metabolic status and response to chemotherapeutic agents in cancer patients, new therapeutic strategies, such as targeting fructose metabolic pathways or immune regulation, are developed to improve the sensitivity and prognosis of cancer patients to chemotherapeutic agents. In conclusion, abnormal fructose metabolism caused by COVID-19 may affect the sensitivity of tumor cells to chemotherapeutic drugs through multiple mechanisms, thus posing new challenges and reflections on clinical treatment strategies.

The influence of SARS-CoV-2 infection on cancer is gradually being explored, and there may be conflicting views [116]. Moreover, the majority of studies focused on lung and blood cancers, minority on gastrointestinal malignancies. Based on a case-control study of 73.4 million cancer patients, including CRC patients, it was concluded that cancer patients are susceptible to COVID-19 infection, and the infection was associated with increased hospitalization and mortality in this population. It is revealed that cancer patients have worse prognosis due to SARS-CoV-2 infection [117]. Our study analyzed the differential expression of fructose metabolism-related genes in CRC patients infected with SARS-CoV-2, and investigated the effect on CRC development. The results provide both experimental data and theoretical basis to further target the regulation of fructose metabolism in CRC patients. Concurrently, these data can also help optimize the dietary guidance and clinical management of CRC patients. Continuous rigorous assessment of outstanding scientific challenges in the laboratory and at the sickbed will address crucial issues, produce new testable hypotheses, but first and foremost improve outcomes for CRC patients with COVID-19. Our study provides a theoretical foundation for the future targeted regulation of fructose metabolism in colorectal cancer patients, while simultaneously optimizing dietary guidance and therapeutic care for colorectal cancer patients in the context of the COVID-19 pandemic.

We made efforts to validate and assure the reliability of the databases used, but we could not entirely eliminate the possibility of inaccuracies due to data restrictions and variations in the data collection procedure. In future investigations, we will increase the data sample size and enhance data validation and analytical procedures to validate our results. We were unable to conduct experimental validation for the new coronavirus due to limitations of objective laboratory conditions and moral issues. Despite utilizing literature research and simulation analysis to draw conclusions, experimental validation is crucial to verify the study's reliability. Future research will require more suitable techniques and resources to conduct wet experiments that can corroborate our conclusions and enhance the credibility and relevance of the study.

## 4. Methods

### Ethics statement

This study has been approved by the China Medical University institutional committee.

### 4.1. Data Collection and Processing

As the largest current cancer gene information database, the TCGA database (https://portal.gdc.cancer.gov/) stores huge amounts of data including gene expression data, miRNA

expression data, copy number variation, DNA methylation, and SNPs. In the study, we downloaded normal (n = 51) and tumor (n = 647) raw mRNA expression data for COADREAD. The NCBI GEO public database Series Matrix File data files for GSE41258 were downloaded using the GPL96 annotation platform, and a total of 185 patient data sets containing complete expression profiles and survival information were retrieved. The GSE17537 Series Matrix File data file was downloaded using the GPL570 annotation platform, and 55 patient data sets with complete expression profiles and survival information were retrieved. GPL24676 was used to download the Series Matrix File data file for GSE183533 (this dataset was exclusively utilized for differential expression analysis). The data include 41 expression profile data sets, which consist of both 31 infected and 10 healthy samples. Using the limma Rpackage package, differential analysis was done, and the differential gene screening conditions were adj.$P$.Val<0.01. This research was conducted using the GeneCards database (http://www.genecards.org/), and a list of 1,431 genes relevant to fructose metabolism was obtained. Through the HMDD database (http://www.cuilab.cn/hmdd), 350 miRNA sets related with colorectal carcinoma were obtained.

## 4.2. Function analysis

Using clusterProfiler, functional annotation of significant gene sets was undertaken to study completely the functional relevance of significant gene sets. Gene Ontology (GO) and Kyoto Encyclopedia of Genes and Genomes (KEGG) were utilized to evaluate the functional categories associated with the genes. Both GO and KEGG enriched pathways with p-values and q-values less than 0.05 were deemed significant. And utilizing the Metascape database (www.metascape.org), functional annotation of significant gene sets was undertaken to study gene sets' functional significance completely. Certain genes were analyzed using GO and KEGG pathways. Min overlap $\geq$ 3 & $P \leq 0.01$ were deemed statistically significant.

## 4.3. Construction and validation of a fructose metabolism-related prognostic model

Using lasso regression, fructose metabolism-related genes with differential expression were chosen to construct a prognostic correlation model for colorectal cancer. Following the incorporation of expression values for each specific gene, a risk score formula was developed for each patient and weighted by the predicted regression coefficients derived from the lasso regression analysis. Using the median risk score value as the dividing line, patients were separated into low-risk and high-risk groups based on the risk score algorithm. Kaplan-Meier and log-rank statistics were used to examine and contrast the survival differences between the two groups. To evaluate the role of the risk score in predicting patient outcome, lasso regression analysis and stratified analysis were utilized. ROC curves were utilized to assess the precision of model predictions.

## 4.4. Immunohistochemistry

Moreover, formalin-fixed paraffin-embedded sample specimens of patients with colorectal cancer were obtained from the Human Protein Atlas Project(https://www.proteinatlas.org). As part of the sample collection supervised by the Uppsala Biobank (http://www.uppsalabiobank.uu.se/en), cancer tissues utilized for protein expression analysis were collected from the Department of Pathology at Uppsala University Hospital, Uppsala, Sweden. Following a microscopic examination of representative, HE sections, cases were selected. From the appropriate tissue blocks, 1 mm-diameter cores were removed and were transferred to cancer tissue microarrays. Sections of microarrays of cancer tissue were immunohistochemically stained

and scanned to generate digital pictures for protein expression analysis. All images were evaluated and were annotated by pathologists in terms of staining intensity and percentage of positive cancer cells for each of the approved antibodies. The immunohistochemistry-based protein expression level was summarized as high, medium, low, or not detected.

### 4.5. Drug sensitivity analysis

We used the Rpackage "pRRophetic" to predict chemosensitivity in each tumor sample based on Genomics of Drug Sensitivity in Cancer (GDSC, https://www.cancerrxgene.org/), which is the largest pharmacogenomics database. IC50 estimates for each specific chemotherapeutic agent treatment were obtained by regression, and 10 cross-validation test regressions and prediction accuracy were performed with the GDSC training set. All parameters were set to their default defaults, which included eliminating "combat" from batch effects and averaging duplicate gene expression. By using regression analysis, IC50 estimations for each specific chemotherapeutic agent treatment were obtained, and regression and prediction accuracy were evaluated using ten cross-validation tests with the GDSC training set. All parameters were set to their default values, which included eliminating "combat" from batch effects and taking the average of duplicate gene expression.

### 4.6. Immune infiltration analysis

To predict the relative proportion of 22 immune infiltrating cells, RNA-seq data from distinct subgroups of COADREAD patients was analyzed using the CIBERSORT method. Risk scores and immune cell content were correlated using Pearson analysis, with $P<0.05$ regarded statistically significant.

### 4.7. Gene Set Variation Analysis (GSVA)

Gene setvariation analysis (GSVA) is a nonparametric, unsupervised method for evaluating the enrichment of transcriptome gene sets. GSVA evaluates the biological function of a sample by exhaustively scoring gene sets of interest in order to turn gene-level changes into pathway-level alterations. In this study, gene sets will be obtained from the Molecular Signatures Database (version 7.0) and the GSVA method will be used to completely score each gene set in order to evaluate potential alterations in biological function between samples.

### 4.8. Gene Set Enrichment Analysis (GSEA)

On the expression profiles of colorectal cancer patients, gene set enrichment analysis (GSEA, http://www.broadinstitute.org/gsea) was done to identify the differently expressed genes between high-risk and low-risk patients. Maximum and minimum gene set sizes of 500 and 15 genes, respectively, were used to filter gene sets. After 100 permutations, abundant gene sets were identified based on a $P < 0.05$ and a false discovery rate (FDR) of 0.20.

### 4.9. Regulatory network analysis of modeled genes

The complex process of transcription initiation in eukaryotes frequently needs a variety of protein factors, which form a transcription initiation complex with RNA polymerase II and engage in the process. In flanking sequences of genes, there are cis-acting elements, including promoters, enhancers, regulatory sequences, and inducible elements, which engage in the regulation of gene expression. Cis-acting elements interact with trans-acting factors to regulate gene expression. This analysis is performed mostly by the Rpackage "RcisTarget", using the Gene-motif rankings database mm9-500 bp-upstream-7species.mc9nr.feather.

### 4.10. Statistical analysis

Using the Kaplan-Meier method, survival curves were created and compared using log-rank. Using the Cox proportional hazards model, multivariate analysis was performed. R was used for every statistical analysis (version 3.6). All statistical tests were two-sided, and $P < 0.05$ was regarded as statistically significant.

## Supporting information

**S1 Table. 333 candidate genes for prognostic associated genes.** Cox univariate regression analysis was used to screen 333 candidate genes for prognostic associated genes.
(PDF)

**S2 Table. All enriched motifs and transcription factors for the modelled genes.**
(PDF)

**S3 Table. 15 mRNA-miRNA relationship pairs.** 15 mRNA-miRNA relationship pairs linked with mRNAs were retrieved from the miRWalk database.
(PDF)

**S4 Table. 2,926 pairs of interactions of lncRNAs.** Interacting lncRNAs were predicted based on miRNAs of S3 Table. A total of 2,926 pairs of interactions were obtained.
(PDF)

**S5 Table. Information in the 3 CRC datasets and the covid dataset.**
(XLSX)

**S1 Code. The relevant code used in this study.** (r).
(R)

## Acknowledgments

We thank the patients and investigators who participated in TCGA, GEO and The Human Protein Atlas Project for providing data.

## Author Contributions

**Data curation:** Jiaxin Jiang.

**Formal analysis:** Jiaxin Jiang.

**Funding acquisition:** Ting Du, Jing Yan.

**Investigation:** Jiaxin Jiang.

**Methodology:** Jiaxin Jiang, Yibo Wang.

**Supervision:** Jing Yan.

**Validation:** Jiaxin Jiang, Jing Yan.

**Writing – original draft:** Jiaxin Jiang, Jing Yan.

**Writing – review & editing:** Jiaxin Jiang, Xiaona Meng, Yibo Wang, Ziqian Zhuang, Ting Du, Jing Yan.

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
