## [Decision Letter · Decision Letter 0]

9 Jan 2024

Dear PhD Yan,

Thank you very much for submitting your manuscript "Effect of aberrant fructose metabolism following SARS-CoV-2 infection on colorectal cancer patients' poor prognosis" for consideration at PLOS Computational Biology.

As with all papers reviewed by the journal, your manuscript was reviewed by members of the editorial board and by several independent reviewers. In light of the reviews (below this email), we would like to invite the resubmission of a significantly-revised version that takes into account the reviewers' comments.

We apologize for the long delay. It was very hard to find expert referees on this topic. We now secured two referee reports that both provide constructive comments.

We cannot make any decision about publication until we have seen the revised manuscript and your response to the reviewers' comments. Your revised manuscript is also likely to be sent to reviewers for further evaluation.

Sincerely,

Rob J De Boer

Section Editor

PLOS Computational Biology

Rob De Boer

Section Editor

PLOS Computational Biology

We apologize for the long delay. It was very hard to find expert referees on this topic. We now secured two referee reports that both provide constructive comments.

We therefore invite you to resubmit a revision addressing these suggestions by the referees.

Reviewer's Responses to Questions

**Comments to the Authors:**

Reviewer #1: SARS-CoV-2-induced cancer patients have a poor prognosis, which may be related to fructose metabolism. In order to improve clinical practice, a composite predictive nomogram is presented in this work. It includes characteristics of the aberrant fructose metabolism brought on by this novel coronavirus, together with age and tumor stage. The genes with the best possible prognostic values were found using LASSO regression analysis. This work lays the theoretical foundation for future customized treatment of fructose metabolism in these patients, while also optimizing dietary advice and therapeutic care for patients with colorectal cancer in the context of the COVID-19 pandemic.

The manuscript is well written. Furthermore, there are other issues that need to be clarified in a revised version.

More detailed comments are provided below.

1. The images are of very poor quality, can’t even understandable. Please re-upload all the images with proper legends and titles.

2. For 1b image, consider the cut-off in log2(FC) so that selected number of genes can be identified properly.

3. Accuracy can not be observed properly as the ROC curve is not clear to view. So that the results can be verified.

4. Here, authors mentioned that they have splitted the dataset into 4:1 ratio for training and validation purpose randomly. Please verify that this ratio performing best or not.

5.Here Lasso regression technique has been implemented. It would be great to compare this regression model with some other techniques? please add one comparison table.

Reviewer #2: This study aims to use a bioinformatics approach to predict fructose metabolism genes that are altered in response to COVID-19 infection and also in tissues from CRC patients, and are linked to CRC prognosis. They developed a prognostic risk score based on a number of these genes, and used this to examine correlations to immune responses and CRC treatment efficacy. The aims of the research have been performed well, there are multiple areas in which the rationale could be explained clearer. The discussion does not discuss any of the results generated in the paper. Therefore, the significance of their findings is not clear, or what their findings will lead to, or how they will be followed up. The outcome of the research is observational, but an interesting hypothesis has been generated.

Major Comments: In most cases, the rationale for performing each type of analysis is not mentioned, there is also no summary of findings in each section which would help link the results to the next analysis. Also, the discussion could be reorganized to discuss the results of the study, it is currently a review of fructose metabolism. There is no mention of any of the findings, or limitations of their work.

Minor Comments:

1. Language in abstract needs improvement

2. The statement “SARS-CoV-2 is more prevalent among cancer patients treated with anticancer therapy than those without anticancer therapy” as a supporting argument needs more thought as this could be due to their increased exposure when not distancing vs anything biological.

3. The supporting citations for the statement the fructose can make cancer worse are for breast and prostate cancers, not CRC. As metabolism differs even by stage and genetic mutations in CRC it is difficult to translate the findings in other cancers to CRC. So this statement should express that limitation.

4. Information on the sex, age, stage and types of tissues analyzed and method of analysis (for example GSE41258 is microarray) in the 3 CRC datasets would be helpful, perhaps as a supplementary table. The lung covid dataset (GSE183533) as generated by RNA-Seq, perhaps this could be mentioned, it’s possible some transcripts are missed in the microarray?

5. Fig 2, It is not clear why the OS time is different between each plot on Figs 2. Typically this is set to 5 years, or in some cases 10 years for CRC, which is the timeframe in which recurrence may occur. This should be more clinically relevant.

6. The rationale for examining immune infiltration was not explained, is this linked to fructose metabolism?

7. Fig 3. A – the risk groups should be labelled in the legend. B, it is not clear which 2 groups are being compared, are these high and low risk groups again?

8. Gender should be changed to sex in text and on figures.

9. Fig 4, what was the rationale for performing univariate and multivariate analysis.

10. There is no information provided on the antibodies used for immunohistochemistry, or their authentication

**Have the authors made all data and (if applicable) computational code underlying the findings in their manuscript fully available?**

Reviewer #1: None

Reviewer #2: **No: **Codes not provided

PLOS authors have the option to publish the peer review history of their article (what does this mean?). If published, this will include your full peer review and any attached files.

Reviewer #1: No

Reviewer #2: No
---

## [Decision Letter · Decision Letter 1]

22 Jul 2024

Dear PhD Yan,

Thank you very much for submitting your manuscript "Effect of aberrant fructose metabolism following SARS-CoV-2 infection on colorectal cancer patients' poor prognosis" for consideration at PLOS Computational Biology. As with all papers reviewed by the journal, your manuscript was reviewed by members of the editorial board and by several independent reviewers. The reviewers appreciated the attention to an important topic. Based on the reviews, we are likely to accept this manuscript for publication, providing that you modify the manuscript according to the review recommendations.

Sincerely,

Rob J De Boer

Section Editor

PLOS Computational Biology

The two referees agree that the manuscript has improved considerably, and have a few more recommendations.

Reviewer's Responses to Questions

**Comments to the Authors:**

Reviewer #1: The authors have addressed my previous comments satisfactorily,

My only remaining comments are:

1. The authors conducted various computational analyses for this study; however, the script for these analyses is not included in the manuscript.

2. The authors have included links to various databases throughout the manuscript. Instead of providing these links, please cite the original papers associated with each database.

Reviewer #2: The authors have been responsive to the recommendations made, however the discussion still mostly discusses research from other studies, and not the results of their own study. The discussion is also very long and given that the research from the paper is not really discussed in any detail, there is room for improvement here.

**Have the authors made all data and (if applicable) computational code underlying the findings in their manuscript fully available?**

Reviewer #1: None

Reviewer #2: None

PLOS authors have the option to publish the peer review history of their article (what does this mean?). If published, this will include your full peer review and any attached files.

Reviewer #1: **Yes: **Amit Ghosh

Reviewer #2: No

Figure Files:

Data Requirements:

Reproducibility:

References:

---

## [Editor Report · Decision Letter 2]

13 Aug 2024

Dear PhD Yan,

We are pleased to inform you that your manuscript 'Effect of aberrant fructose metabolism following SARS-CoV-2 infection on colorectal cancer patients' poor prognosis' has been provisionally accepted for publication in PLOS Computational Biology.

Best regards,

Rob J De Boer

Section Editor

PLOS Computational Biology

---

## [Editor Report · Acceptance letter]

23 Sep 2024

PCOMPBIOL-D-23-00881R2 

Effect of aberrant fructose metabolism following SARS-CoV-2 infection on colorectal cancer patients' poor prognosis

Dear Dr Yan,

I am pleased to inform you that your manuscript has been formally accepted for publication in PLOS Computational Biology. Your manuscript is now with our production department and you will be notified of the publication date in due course.

With kind regards,

Zsofia Freund
